# Short-Term Outcome of Robotic versus Laparoscopic Hysterectomy for Endometrial Cancer in Women with Diabetes: Analysis of the US Nationwide Inpatient Sample

**DOI:** 10.3390/jcm12247713

**Published:** 2023-12-15

**Authors:** Huang-Pin Shen, Chih-Jen Tseng

**Affiliations:** 1Institute of Medicine, Chung Shan Medical University, Taichung 40201, Taiwan; 2Department of Obstetrics and Gynecology, Chung Shan Medical University Hospital, Taichung 40201, Taiwan; 3School of Medicine, Chung Shan Medical University, Taichung 40201, Taiwan

**Keywords:** diabetes, endometrial cancer (EC), hysterectomy, in-hospital outcome, Nationwide Inpatient Sample (NIS)

## Abstract

This study investigated short-term outcomes of robotic versus laparoscopic hysterectomy for endometrial cancer (EC) in women with diabetes. We extracted the data of hospitalized females aged ≥18 years who were diagnosed with EC and diabetes and underwent robotic or laparoscopic hysterectomy from the US Nationwide Inpatient Sample (NIS) 2005–2018. Associations between study variables and in-hospital outcomes, including complications, unfavorable discharge, length of stay (LOS), and hospital costs, were examined using logistic regression. A total of 5745 women (representing 28,176 women in the US) were included. Multivariable analysis revealed that robotic surgery was significantly associated with a decreased risk of unfavorable discharge (adjusted odds ratio [aOR] = 0.63, 95% confidence interval [CI]: 0.46, 0.85) than pure laparoscopic surgery. Women who underwent robotic surgery had a significantly shorter LOS (0.46 fewer days, 95% CI: −0.57, −0.35) but higher total hospital costs (6129.93 greater USD; 95% CI: 4448.74, 7811.12). Compared with pure laparoscopic surgery, robotic hysterectomy was associated with less unfavorable discharge among women aged ≥60 years (aOR = 0.60, 95% CI: 0.44, 0.80). For US women with EC and diabetes, robotic hysterectomy is associated with shorter LOS, decreased risk of unfavorable discharge, especially among older patients, and higher total costs than laparoscopic surgery.

## 1. Introduction

Endometrial cancer (EC) is the most prevalent gynecologic cancer and ranks fourth in frequency among women in the United States (US) [1]. Incidence rates for EC have risen consistently over several decades, increasing from 23 per 100,000 individuals in 1990 to 27 per 100,000 individuals in 2017 [2]. Mortality rates have exhibited a similar trend during the same period [3]. Surgical intervention, primarily encompassing hysterectomy, bilateral salpingo-oophorectomy, and lymph node assessment, remains the cornerstone of management for the majority of EC patients, often followed by adjuvant radiation and chemotherapy [4].

Various surgical approaches, including vaginal hysterectomy, laparoscopic hysterectomy, robotic-assisted laparoscopic hysterectomy, and abdominal hysterectomy, are available for performing hysterectomy [5,6]. Among these approaches, laparoscopic hysterectomy and robotic-assisted laparoscopic hysterectomy involve smaller abdominal incisions and are currently favored. Compared with open surgeries, these minimally invasive procedures offer advantages such as reduced operative time, shorter hospital stays, and improved cosmetic outcomes [7].

Diabetes has long been linked to various cancers [8]. A comprehensive review by Saed et al. (2019) [9] evaluated 22 cohort and case-control studies that investigated potential associations between diabetes and the development of EC. The findings of that study were consistent with an earlier review and meta-analysis (2007), showing that diabetes was associated with a 72% increased risk for EC [9,10]. Another larger population-based study emphasized that women with EC and type 2 diabetes mellitus (T2DM) had a twofold higher overall mortality rate, cancer-specific mortality rate, and recurrence rate compared to those without T2DM [11]. In addition, diabetes is also a well-known adverse prognostic factor for short-term outcomes after most surgical procedures, increasing the risk for postoperative complications and reoperations [12,13].

Several prior studies have compared robotic and laparoscopic hysterectomy for EC among diverse populations [14,15,16]. A study by Eoh et al. suggested that robotic hysterectomy is a safe surgical alternative to laparoscopic hysterectomy, especially for women with low-risk EC. Surgical and oncologic results are equivalent between the two procedures [16]. Another study investigated older women with EC, demonstrating that robotic surgery had lower blood loss but a longer operative time than laparoscopic hysterectomy [15]. Yet another study compared these two procedures in obese women and reported a similar rate of perioperative complications [14]. However, to date, no study has focused on possible differences in the outcomes between these two procedures among women with diabetes.

Given the existing knowledge regarding EC, diabetes, and minimally invasive hysterectomy, it becomes necessary to examine whether laparoscopic and robotic hysterectomy yield distinct short-term outcomes for women with EC and diabetes. This exploration is particularly essential since diabetic patients often present with additional challenges in surgical settings, such as elevated chances of complications and higher medical expenses, as mentioned previously. Therefore, this study evaluated and compared the short-term outcomes, including postoperative complications, length of stay (LOS), and total hospital costs between robotic and pure laparoscopic hysterectomy in women with EC and diabetes, using a comprehensive nationwide inpatient dataset. 

## 2. Materials and Methods

### 2.1. Study Design and Data Source

This retrospective, population-based observational study extracted all data from the US Nationwide Inpatient Sample (NIS) database, which is the largest all-payer, continuous inpatient care database in the United States, including about 8 million hospital stays each year [17]. The database is administered by the Healthcare Cost and Utilization Project (HCUP) of the US National Institutes of Health (NIH). Patient data include primary and secondary diagnoses, primary and secondary procedures, admission and discharge status, patient demographics, expected payment source, duration of hospital stay, and hospital characteristics (i.e., bed size/location/teaching status/hospital region). All admitted patients are initially considered for inclusion. The continuous, annually updated NIS database derives patient data from about 1050 hospitals from 44 States in the US, representing a 20% stratified sample of US community hospitals as defined by the American Hospital Association.

### 2.2. Ethics Statement

This study complies with the terms of the NIS data-use agreement (certification code: HCUP-894EWT39I). The data utilized in this study were obtained through the Online HCUP Central Distributor. Given that this study solely involved the analysis of secondary data, there was no direct involvement of the general public or patients. It was granted an exemption from requiring IRB approval.

### 2.3. Study Population

Hospitalized female patients ≥18 years old with EC and diabetes who were undergoing robot-assisted or pure laparoscopic hysterectomy were included in this study. Patients with missing information on main endpoints, lymph node invasion, and metastatic disease were excluded from the cohort. All diagnoses and procedures were identified by corresponding International Classification of Diseases, Ninth and Tenth edition (ICD-9 and ICD-10) diagnostic and procedure codes.

### 2.4. Study Variables and Outcome Measures

Primary outcomes were short-term outcomes, including (1) length of stay (LOS), (2) unfavorable discharge, defined as discharge to nursing homes or long-term care facilities, (3) total hospital costs, and (4) postoperative complications. 

### 2.5. Postoperative Complications 

Postoperative complications included acute myocardial infarction (AMI), cerebrovascular accident (CVA), venous thromboembolism (VTE), pneumonia, sepsis, surgical site infection, major blood loss, respiratory failure, mechanical ventilation, wound dehiscence, acute kidney injury, urinary tract infection, and in-hospital death. The detailed ICD codes for identifying the said conditions are summarized in Appendix A.

### 2.6. Other Variables

Patients’ characteristics included age (continuous and by category), race, insurance status (primary payer), household income, smoking, obesity (defined as BMI ≥ 30 kg/m^2^), admission year, weekend admission, major comorbidities, emergency admission, and Charlson comorbidity index (CCI). Hospital-related characteristics such as bed size, location/teaching status, and hospital region were extracted from the database as part of the comprehensive data available for all participants. Major comorbidities included chronic kidney disease (CKD), ischemic heart disease, congestive heart failure, atrial fibrillation, anemia, chronic obstructive pulmonary disease (COPD), cerebrovascular disease, peripheral vascular disease, severe liver disease, rheumatic disease, and coagulopathy. The detailed ICD codes for defining the comorbidities and complications are documented in Appendix A.

### 2.7. Statistical Analysis

Descriptive statistics of the patients are presented as unweighted counts (*n*) and weighted percentage (%) or mean ± standard error (SE). Since the NIS database covers 20% of samples of the USA annual inpatient admissions, weighted samples (before 2011 using TrendWT and after 2012 using DISCWT), stratum (NIS_STRATUM), cluster (HOSPID) were used to produce national estimates for all analyses. The *p*-values for group comparisons were computed using PROC SURVEYFREQ for categorical data and SURVEYREG for continuous data. The SURVEYFREQ procedure provides a Rao–Scott chi-square test to test the significance between weighted proportions [18]. The SURVEYREG procedure fits linear models for survey data and provides significance tests for the model effects [19]. Logistic regression models were performed using PROC SURVEYLOGISTIC, and linear regression models were performed using SURVEYREG to identify the associations between study variables and risk for (any) complication, unfavorable discharge, LOS, and hospital costs in patients undergoing different types of hysterectomy surgery. Multivariable regression was adjusted for variables that were significant (*p* < 0.05) in univariate analysis (except for CCI), as shown in Table 1, including age (categorical), race, insurance status, household income, smoking, obesity, study year, hospital region, hospital location/teaching status, CKD, peripheral vascular disease, diabetes with chronic complication, and emergency admission. Stratified analysis was employed to identify the risk for (any) complication, unfavorable discharge, LOS, and hospital costs in patients undergoing different types of hysterectomy surgery. All *p* values were two-sided, and *p* < 0.05 was considered statistically significant. All statistical analyses were performed using the statistical software package SAS software version 9.4 (SAS Institute Inc., Cary, NC, USA).

## 3. Results

### 3.1. Study Population Selection

The study population selection process is depicted in Figure 1. A total of 6318 hospitalized females aged ≥18 years with EC and diabetes, who were undergoing robot-assisted or pure laparoscopic hysterectomy in the 2005–2018 dataset were included. Of these, 105 women with missing information on study outcomes were excluded. Another 462 patients with lymph node invasion or metastatic disease and 6 patients without weight values of the dataset were excluded. Finally, the data of 5745 women remained for subsequent analyses. This sample represents 28,176 hospitalized women in the entire U.S. after weighting (Figure 1).

### 3.2. Characteristics of Women with Diabetes Undergoing Robotic or Pure Laparoscopic Hysterectomy for EC

Patients’ demographics, major comorbidities, and hospital-related information are summarized in Table 1. Among the study population, 3668 women received robotic hysterectomy, while 2077 underwent a pure laparoscopic procedure. The mean age of all women was 64.2 years. Of these, 3691 women (69.9%) were white, and 3462 (60.4%) had insurance covered by Medicare/Medicaid. The most common comorbidity was COPD (13.1%).

### 3.3. In-Hospital Outcomes of Women with Diabetes Undergoing Robotic or Pure Laparoscopic Hysterectomy for EC

In-hospital outcomes of women with diabetes undergoing minimally invasive hysterectomy for EC are summarized in Table 2. The robot-assisted group had a shorter mean LOS than the laparoscopic group (2.19 vs. 2.57 days). The frequency of postoperative complications was not significantly different between the two procedures (18.8% vs. 17.4%). Total hospital costs were significantly higher in the robotic surgery group compared to those in the laparoscopic group (58,055.8 vs. 44,253.8 USD).

### 3.4. Associations between the Types of Minimally Invasive Hysterectomy and In-Hospital Outcomes

Associations between the type of minimally invasive hysterectomy and in-hospital outcomes in women with diabetes are summarized in Table 3. After adjusting for relevant confounders, including age, race, insurance status, household income, smoking, obesity, study year, hospital region, hospital location/teaching status, CKD, peripheral vascular disease, diabetes with chronic complications, and emergency admission, multivariable analysis revealed that robotic surgery was significantly associated with a decreased risk of unfavorable discharge (adjusted odds ratio [aOR] = 0.63, 95% confidence interval [20]: 0.46, 0.85) compared with that of pure laparoscopic surgery. Also, women who underwent robotic surgery had a significantly shorter LOS (0.46 fewer days, 95% CI: −0.57, −0.35) yet still had greater total hospital costs (6129.93 greater USD; 95% CI: 4448.74, 7811.12) than those associated with pure laparoscopic surgery. The full statistical model for the associations between study variables and in-hospital outcomes is shown in Appendix A.

### 3.5. Associations between Types of Minimally Invasive Hysterectomy and In-Hospital Outcomes, Stratified by Age and Obesity Status

The results of analyses stratified by age and obesity status are shown in Table 4. In women aged > 60 years, those undergoing robotic surgery had significantly lower odds for unfavorable discharge (aOR = 0.60, 95% CI: 0.44, 0.80) than pure laparoscopic surgery after adjusting for confounders.

As compared with pure laparoscopic surgery, robotic hysterectomy was significantly associated with lower odds for unfavorable discharge among women aged >60 years (aOR = 0.60, 95% CI: 0.44, 0.80) but not among those aged <60 years. In addition, robotic surgery was associated with lower odds for unfavorable discharge regardless of obesity status (obese, aOR = 0.65, 95% CI: 0.43, 0.98; non-obese, aOR = 0.65, 95% CI: 0.41, 0.94).

Robotic surgery was significantly associated with shorter LOS than pure laparoscopic surgery regardless of age and obesity status (age <60 years: −0.35 days, 95% CI: −0.43, −0.26; age >60 years: −0.50 days, 95% CI: −0.65, −0.35; obese: −0.42 days, 95% CI: −0.54, −0.30; non-obese, −0.45 days, 95% CI: −0.62, −0.28). However, compared to pure laparoscopic hysterectomy, robotic surgery was not significantly associated with the risk of (any) postoperative complications.

## 4. Discussion

### 4.1. Summary of Main Results

In the present study, among diabetic women with EC, robotic hysterectomy was associated with a significantly reduced risk for unfavorable discharge compared to pure laparoscopic hysterectomy, particularly in women older than age 60 years. The hospital LOS was also nearly half a day shorter for robotic surgery than for pure laparoscopic surgery. The stratified analysis revealed that these associations were not modified by the presence of obesity. However, the robotic hysterectomy procedure was also associated with higher total hospital costs than pure laparoscopic hysterectomy in women with diabetes and EC.

### 4.2. Results in the Context of Published Literature

This study compared the minimally invasive treatments for EC in patients who had comorbid diabetes. Diabetes and cancer, both the risk of acquiring cancer and dying from cancer, have been widely studied in the literature in relation to various cancers. An umbrella review assessed associations between cancer at 20 sites and mortality of cancer at seven sites, reporting that increased risk was shown at all sites compared to being without diabetes [21]. Regarding EC specifically, a prior systematic review and meta-analysis conducted earlier in 2007 revealed a favorable correlation between diabetes and the likelihood of developing EC [22]. That study attributed the main contributors to type 2 diabetes mellitus (T2DM) as insulin resistance/hyperinsulinemia and low levels of physical activity/obesity, all of which are understandably recognized risk factors for EC. Another study whose authors agreed that excess circulating insulin—the hyperinsulinemia pathway—contributed to EC development confirmed that endometrial hyperplasia and subsequent EC were significantly associated with a history of gestational diabetes (relative to insulin resistance in pregnancy), specifically in younger women [23]. Ten years later, an updated review and meta-analysis [9] demonstrated significant associations supporting the fact that women with diabetes had an increased risk of EC. Moreover, diabetes is also a well-known adverse prognostic factor for short-term surgical outcomes, with an increased risk for postoperative complications and reoperations in various surgical settings [12,13]. Motivated by such results, the present study is the first to focus solely on the possible impact on the short-term outcomes of different types of minimally invasive hysterectomy among women with diabetes.

Notably, in the present study, robotic hysterectomy was associated with significantly shorter hospital LOS than that of pure laparoscopic hysterectomy, regardless of age and obesity status. Inpatient stays are already reduced for women undergoing minimally invasive hysterectomies compared to the outcomes of open hysterectomies [16]. Robotic surgery is considered to be less physically demanding and fatiguing for surgeons, leading to improved focus and performance. Additionally, the use of low CO_2_ pressure during robotic surgery does not adversely affect the visualization of the surgical field. This reduction in CO_2_ pressure has several benefits, including a decrease in anesthesia requirements and associated complications, ultimately creating a more favorable peritoneal environment for the procedure [20,24]. These characteristics of the robot-assisted procedure thus explain the improvement in certain patient outcomes and recovery postoperatively, especially benefitting older patients and those with chronic comorbidities such as patients with diabetes.

According to our findings, for diabetic women over the age of 60, the robot-assisted procedure appears to offer a significant benefit by reducing the risk of unfavorable discharge and possibly avoiding the need for long-term care, as compared to pure laparoscopic procedures. The precision and dexterity inherent in robotic-assisted procedures enable improved visualization, ensuring surgeries are more precise. This advanced technology potentially hastens postoperative recovery and thereby reduces the risk of discharge to long-term care facilities. However, the same favorable outcome was not observed for younger diabetic women. Younger diabetic women may already have a lower risk of complications due to their physical reserve and the less advanced stage of EC. In such cases, the benefits of robotic surgery may not be as pronounced, as these individuals may already have a generally favorable prognosis.

In the literature, the transition from open surgery to robotic surgery for early stage endometrial cancer (EC) in Denmark was linked to noteworthy decreases in major complications, including a significant reduction in 30-day mortality [25]. In the present study, the number of in-hospital deaths was remarkably low, with only 13 cases recorded, comprising seven in the robotic group and six in the laparoscopic group. However, it is important to note that our analysis could not compare the long-term oncologic outcomes between robotic and pure laparoscopic procedures after discharge due to a lack of available data in this regard. Nevertheless, a meta-analysis of six cohort studies provided inconclusive evidence for establishing a definitive link between diabetes and EC-specific mortality [26]. More recent studies have indicated a two-fold increase in EC-specific mortality among patients with type 2 diabetes mellitus (T2DM) compared to those without [11,27].

Diabetic states such as hyper- and hypoglycemia and high glycemic variability are known to aggravate patients’ disease states, increasing the likelihood of adverse events associated with surgery [12,18]. However, robotic surgery seemed not to offer more advantages in reducing short-term post-surgical complications over pure laparoscopic procedures in our analysis. Last but not least, it is demonstrated that the total hospital costs of robot-assisted surgery were higher than those for pure laparoscopic surgery in our study. This aligns with general expectations since robotic surgery tends to be more expensive than open surgery and other minimally invasive surgeries. However, some researchers have observed that the cost differences become less significant when considering the shorter hospital stay and reduced complications associated with robotic surgery [28]. Nonetheless, it is important to carefully consider and weigh the advantages of robot-assisted procedures compared to pure laparoscopic procedures, taking into account their associated benefits and costs.

### 4.3. Strengths and Weaknesses

One of the major strengths of the present study is the inclusion of a relatively large in-patient sample with comprehensive data on hospitalization, allowing us to study various perioperative events, even those rarely seen in clinical practice. Nevertheless, the present study is inherently constrained by its retrospective and observational design, which restricts the general applicability of the findings to other populations and prevents the establishment of causal relationships. Selection bias also cannot be ruled out in population-based retrospective studies. Like in other reports utilizing ICD code systems, the possibility for coding errors also exists in this analysis. In addition, the stages of EC among the patients were unknown due to a lack of data. Information on factors such as baseline performance status, node dissection, or prophylactic antibiotics usage was lacking and thus could not be analyzed. The NIS also does not provide information about intraoperative parameters such as surgeon’s experience, duration of operation, and exact blood loss. Although we have carefully adjusted for the relevant confounders in the multivariable analysis, factors that were not included or collected by the dataset could still bias the results. While vital for individual cases, the assessment of long-term outcomes after discharge was not possible due to the limitation of NIS data, which solely encompasses in-patient information.

## 5. Conclusions

Among minimally invasive hysterectomy procedures, robot-assisted hysterectomy significantly reduces the risk of unfavorable discharge and shortens hospital stays for US women with EC and diabetes, especially older women, when compared to pure laparoscopic hysterectomy. Nevertheless, the total hospital costs associated with robotic hysterectomy are higher than those of pure laparoscopic surgery.

## Figures and Tables

**Figure 1 jcm-12-07713-f001:**
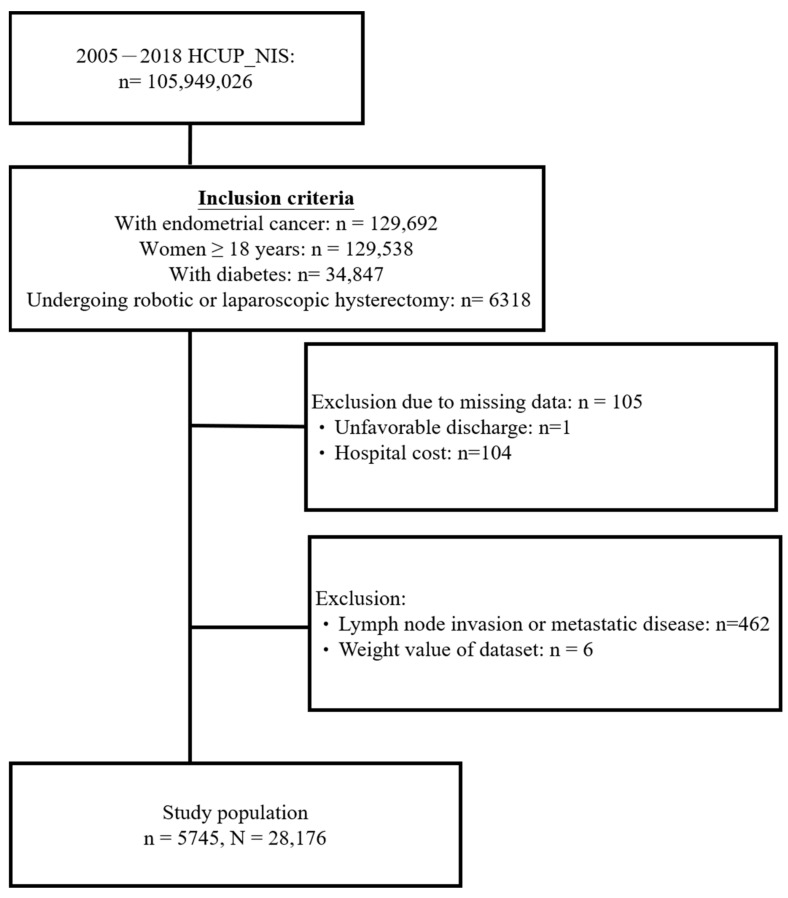
Flow diagram of study cohort selection.

**Table 1 jcm-12-07713-t001:** Baseline characteristics of women with diabetes undergoing minimally invasive hysterectomy for endometrial cancer.

Characteristics	Total (*n* = 5745)	Robotic (*n* = 3668)	Pure Laparoscopic (*n* = 2077)	*p*-Value
Demography				
Age, years	64.2 ± 0.15	64.3 ± 0.18	64.1 ± 0.24	0.374
18–49	482 (8.4)	279 (7.6)	203 (9.8)	**0.003**
50–59	1347 (23.4)	854 (23.2)	493 (23.8)	
60–69	2146 (37.4)	1418 (38.7)	728 (35.1)	
70–79	1297 (22.6)	833 (22.7)	464 (22.3)	
≥80	473 (8.2)	284 (7.7)	189 (9.0)	
Race				
White	3691 (69.9)	2454 (71.4)	1237 (67.2)	**0.003**
Black	572 (10.9)	372 (10.8)	200 (10.9)	
Hispanic	608 (11.6)	384 (11.3)	224 (12.3)	
Other	399 (7.6)	224 (6.5)	175 (9.6)	
Missing	475	234	241	
Insurance status				
Medicare/Medicaid	3462 (60.4)	2244 (61.2)	1218 (58.8)	**<0.001**
Private including HMO	1985 (34.6)	1278 (34.9)	707 (34.0)	
Self-pay/no-charge/other	291 (5.1)	143 (3.9)	148 (7.2)	
Missing	7	3	4	
Household income				
Q1	1475 (26.2)	911 (25.3)	564 (28.0)	**<0.001**
Q2	1417 (25.1)	947 (26.2)	470 (23.1)	
Q3	1426 (25.3)	949 (26.4)	477 (23.4)	
Q4	1319 (23.3)	801 (22.1)	518 (25.6)	
Missing	108	60	48	
Smoking	1041 (18.2)	727 (19.9)	314 (15.2)	**<0.001**
Obesity	2877 (50.3)	2014 (55.0)	863 (41.9)	**<0.001**
Study years (NIS dataset)				
2005–2009	1132 (19.2)	235 (6.3)	897 (42.4)	**<0.001**
2010–2015	3401 (59.3)	2519 (68.4)	882 (42.8)	
2016–2018	1212 (21.5)	914 (25.2)	298 (14.8)	
Weekend admission	154 (2.7)	106 (2.9)	48 (2.3)	0.130
Hospital bed size				
Small	533 (9.1)	313 (8.4)	220 (10.2)	0.229
Medium	1192 (21.0)	744 (20.6)	448 (21.6)	
Large	3978 (70.0)	2577 (71.0)	1401 (68.2)	
Missing	42	34	8	
Hospital region				
Northeast	1339 (23.5)	766 (20.9)	573 (28.1)	**<0.001**
South	1264 (21.9)	935 (25.3)	329 (15.9)	
Midwest	1799 (31.1)	1148 (31.2)	651 (30.9)	
West	1343 (23.5)	819 (22.6)	524 (25.1)	
Hospital location/teaching status				
Rural	101 (1.7)	31 (0.8)	70 (3.4)	**<0.001**
Urban nonteaching	1085 (18.7)	629 (17.1)	456 (21.4)	
Urban teaching	4517 (79.6)	2974 (82.0)	1543 (75.2)	
Missing	42	34	8	
Major comorbidities				
CKD	234 (4.1)	177 (4.9)	57 (2.8)	**<0.001**
Ischemic heart disease	736 (12.8)	481 (13.1)	255 (12.2)	0.289
Congestive heart failure	364 (6.3)	233 (6.4)	131 (6.3)	0.934
Atrial fibrillation	431 (7.5)	282 (7.7)	149 (7.1)	0.449
Anemia	540 (9.4)	351 (9.6)	189 (9.1)	0.545
COPD	752 (13.1)	488 (13.2)	264 (12.7)	0.571
Cerebrovascular disease	138 (2.4)	94 (2.6)	44 (2.1)	0.229
Peripheral vascular disease	132 (2.3)	95 (2.6)	37 (1.8)	**0.042**
Severe liver disease	30 (0.5)	23 (0.6)	7 (0.3)	0.141
Rheumatic disease	101 (1.8)	66 (1.8)	35 (1.7)	0.752
Coagulopathy	105 (1.8)	72 (2.0)	33 (1.6)	0.268
Diabetes with chronic complications	537 (9.4)	400 (11.0)	137 (6.7)	**<0.001**
Emergency admission	626 (10.9)	353 (9.7)	273 (13.2)	**0.001**
CCI				
0	3821 (66.4)	2380 (64.8)	1441 (69.4)	**<0.001**
1	1102 (19.2)	717 (19.5)	385 (18.5)	
2+	822 (14.4)	571 (15.6)	251 (12.1)	

CKD—chronic kidney disease; COPD—chronic obstructive pulmonary disease; CCI—Charlson comorbidity index; HMO—Health Maintenance Organization; Q—quartile. Continuous variables are presented as mean ± SE; categorical variables are presented as unweighted counts (weighted percentage). Significant difference with *p*-values < 0.05 are shown in bold.

**Table 2 jcm-12-07713-t002:** In-hospital outcomes of women with diabetes undergoing minimally invasive hysterectomy for endometrial cancer.

	Total (*n* = 5745)	Robotic (*n* = 3668)	Pure Laparoscopic (*n* = 2077)	*p*-Value
Complications, any	1050 (18.3)	688 (18.8)	362 (17.4)	0.203
In-hospital death	13 (0.2)	7 (0.2)	6 (0.3)	0.476
AMI	22 (0.4)	11 (0.3)	11 (0.5)	0.201
CVA	56 (1.0)	43 (1.2)	13 (0.6)	**0.021**
VTE	44 (0.8)	23 (0.6)	21 (1.0)	0.094
Pneumonia	49 (0.9)	30 (0.8)	19 (0.9)	0.660
Sepsis	85 (1.5)	54 (1.5)	31 (1.5)	0.928
Surgical site infection	21 (0.4)	16 (0.4)	5 (0.2)	0.239
Major blood loss	577 (10.0)	358 (9.8)	219 (10.5)	0.377
Respiratory failure/mechanical ventilation	265 (4.6)	177 (4.8)	88 (4.2)	0.270
Wound dehiscence	25 (0.4)	13 (0.4)	12 (0.6)	0.223
AKI	243 (4.3)	176 (4.8)	67 (3.2)	**0.004**
UTI	41 (0.7)	25 (0.7)	16 (0.8)	0.688
LOS ^a^	2.32 ± 0.05	2.19 ± 0.04	2.57 ± 0.10	**<0.001**
Unfavorable discharge ^a^	278 (4.9)	165 (4.6)	113 (5.5)	0.103
Total hospital costs	53,120.7 ± 799.0	58,055.8 ± 915.3	44,253.8 ± 1255.5	**<0.001**

AMI–acute myocardial infarction; CVA–cerebrovascular accident; VTE–venous thromboembolism; AKI–acute kidney injury; UTI–urinary tract infection; CKD–chronic kidney disease; COPD–chronic obstruction pulmonary disease; CCI–Charlson comorbidity index. Continuous variables are presented as mean ± SE; categorical variables are presented as unweighted counts (weighted percentage). Significant difference with *p*-values < 0.05 are shown in bold. ^a^ Excluded patients with in-hospital mortality.

**Table 3 jcm-12-07713-t003:** Associations between types of minimally invasive hysterectomy (robotic vs. pure laparoscopic) and in-hospital outcomes in women with diabetes. (*n* = 5123).

	Complication, Any	Unfavorable Discharge ^a^	LOS ^a^	Total Hospital Costs, US Dollars
	aOR ^b^ (95% CI)	aOR ^b^ (95% CI)	aBeta ^b^ (95% CI)	aBeta ^b^ (95% CI)
Pure laparoscopic	Ref.	Ref.	Ref.	Ref.
Robot-assisted	0.88 (0.75, 1.04)	**0.63 (0.46, 0.85)**	**−0.46 (−0.57, −0.35)**	**6129.93 (4448.74, 7811.12)**

aOR—adjusted odds ratio; CI—confidence interval; ref.—reference. Variables with *p*-values < 0.05 are shown in bold. ^a^ Excluded patients with in-hospital mortality. ^b^ Adjusted for variables with a *p*-value < 0.05 in Table 1 (except for CCI), including age, race, insurance status, household income, smoking, obesity, study year, hospital region, hospital location/teaching status, CKD, peripheral vascular disease, diabetes with chronic complications, and emergency admission.

**Table 4 jcm-12-07713-t004:** Associations between minimally invasive hysterectomy (robotic vs. pure laparoscopic) and in-hospital outcomes in women with diabetes, stratified by age and obesity status.

	Number of Women ^b^	Complication, Any	Unfavorable Discharge ^a^	LOS ^a^
		aOR ^c^ (95% CI)	aOR ^c^ (95% CI)	aBeta ^c^ (95% CI)
Age				
<60	1617	0.92 (0.66, 1.29)	0.93 (0.27, 3.15)	**−0.35 (−0.43, −0.26)**
>60	3506	0.87 (0.72, 1.05)	**0.60 (0.44, 0.80)**	**−0.50 (−0.65, −0.35)**
Obesity status				
Obese	3587	0.88 (0.71, 1.11)	**0.65 (0.43, 0.98)**	**−0.42 (−0.54, −0.30)**
Non-obese	2536	0.92 (0.73, 1.16)	**0.62 (0.41, 0.94)**	**−0.45 (−0.62, −0.28)**

aOR—adjusted odds ratio; CI—confidence interval; LOS—length of stay. Variables with *p*-values < 0.05 are shown in bold. ^a^ Excluded patients with in-hospital mortality. ^b^ Excluded patients with missing data on the adjusted variables. ^c^ Adjusted for variables with a *p*-value < 0.05 in Table 1 (except for CCI and the stratified variable), including age, race, insurance status, household income, smoking, obesity, study year, hospital region, hospital location/teaching status, CKD, peripheral vascular disease, diabetes with chronic complications, and emergency admission.

## Data Availability

Data are contained within the article.

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
