# Peer review of "Short-Term Outcome of Robotic versus Laparoscopic Hysterectomy for Endometrial Cancer in Women with Diabetes: Analysis of the US Nationwide Inpatient Sample"

_jcm, 2023, doi:10.3390/jcm12247713_

Round 1
Reviewer 1 Report
Comments and Suggestions for Authors
Problem of the role of robotic surgery vs. traditional laparoscopy in endometrial cancer is morden nowadays. The topic is relevant in the field. Unfortunately work doesn't add to the subject area new knowledge compared with other published documents. Authors retrospectively compared results of treatment by this two methods in patients with diabetes. There is no specific theoretical reason for, why the patients with diabetes would have a different results than others in that compersion of two surgical techniques. I suggest authors schuld explain why the patients with diabetes were supposed to have different result then others patients, when undergo robotic or laparoscopic procedure for endometrial cancer. The conclusions consistent with the evidence and arguments presented and they address the main question posed.
The references and the tables are appropriate
Discussion section should be improved by possible explanation: why robotic hysterectomy is associated with decreased risk of unfavorable discharge. This doesn't seem to be understable yet.
Author Response
The abstract section has been amended to a total of 200 words, following the style of structured abstracts. Reviewer 1 Problem of the role of robotic surgery vs. traditional laparoscopy in endometrial cancer is morden nowadays. The topic is relevant in the field. Unfortunately, work doesn't add to the subject area new knowledge compared with other published documents. Authors retrospectively compared results of treatment by these two methods in patients with diabetes. There is no specific theoretical reason for, why the patients with diabetes would have a different result than others in that comparison of two surgical techniques. Author Response: We appreciate the thoughtful comments and acknowledge the importance of the role of robotic surgery versus traditional laparoscopy in endometrial cancer. Our primary aim was to specifically investigate and compare the short-term outcomes of robotic surgery and traditional laparoscopy in endometrial cancer patients with diabetes. While it is true that some studies have compared these surgical techniques in general, our study adds value by honing in on a specific subgroup – those with diabetes. This subgroup is of particular interest due to the unique challenges posed by diabetes in the various surgical setting and its known association with adverse outcomes. I suggest authors should explain why the patients with diabetes were supposed to have different result then others patients, when undergo robotic or laparoscopic procedure for endometrial cancer. Author Response: We appreciate the thoughtful comment and acknowledge the importance of providing clarity on why patients with diabetes were expected to have different outcomes when undergoing robotic or laparoscopic procedures for endometrial cancer. The rationale for this lies in the association between diabetes and various adverse health outcomes, both in the context of cancer and surgical procedures. Introduction section is amended to better reflect the rationale. The conclusions consistent with the evidence and arguments presented and they address the main question posed. The references and the tables are appropriate Author Response: Thank you very much for your comment. Discussion section should be improved by possible explanation: why robotic hysterectomy is associated with decreased risk of unfavorable discharge. This doesn't seem to be understable yet. Author Response: We appreciate the comment regarding the association between robotic hysterectomy and a decreased risk of unfavorable discharge. In response, we have revised the Discussion section to provide a clearer explanation. Please check the revised Discussion section.Reviewer 2 Report
Comments and Suggestions for Authors
Review reports
· A brief summary
- This study aimed to compare the short-term outcome of 2 minimally invasive surgical (MIS) procedures among endometrial cancer patients with DM by using the US Nationwide Inpatient Sample (NIS) database 2005-2018. The result showed no significant differences were observed in postoperative complications between the two procedures. However, robotic hysterectomy is associated with shorter LOS, decreased risk of unfavorable discharge, especially among older patients, and higher total hospital costs, than pure laparoscopic surgery.
- General concept comments
Title: Due to the main outcome is the short-term outcome of 2 MIS techniques. The authors should be more specific in the title linked to this
“ The short-term outcome of robotic versus laparoscopic hysterectomy for endometrial cancer in women with diabetes: Analysis of the US Nationwide Inpatient Sample …….”
Abstract:
Due to the unbalanced basic characteristics of these 2 participants in the robotic and laparoscopic procedure and unclear stat that using in this study, the result might be bias
Introduction
Please give more specific main outcome that aimed to compare the short term outcome of these 2 MIS procedures in this part
Methods
Please give more detail about the stat used in this study.
What about the patients aged equal to 60
Please give more detail about the other comorbidity disease in these 2 group of study patients other than DM
Please give more detail about the surgical procedure other than hysterectomy such as pelvic and/or paraaortic node dissection
No detail about the preop prophylaxis antibiotic in these 2 MIS procedures
What was the definition of obesity in this study?
Results
Why exclude the patients with lymph node invasion? Please add this exclusion criterion in the method part
Page 6: The authors mentioned about after adjusted for relevant confounder …, please give more detail about the variable factors
Discussion
Summary of the main results
Due to the unbalanced basic characteristics of these 2 participants in the robotic and laparoscopic procedure and the unclear stat that is used in this study, the result might be biased.
Table 1 : please specify the stat used to compare in this table (Chi-squared???)
Table 3: please give more detail about the variables that were adjusted. Which statistic was used?
Table 4: What about the patients aged equal to 60?
Comments on the Quality of English Language
Minor editing of English language required
Author Response
• General concept comments Title: Due to the main outcome is the short-term outcome of 2 MIS techniques. The authors should be more specific in the title linked to this “ The short-term outcome of robotic versus laparoscopic hysterectomy for endometrial cancer in women with diabetes: Analysis of the US Nationwide Inpatient Sample …….” Author Response: The title has been revised accordingly. Abstract: Due to the unbalanced basic characteristics of these 2 participants in the robotic and laparoscopic procedure and unclear stat that using in this study, the result might be bias Author Response: Baseline characteristics of the two groups were unbalanced, but we meticulously adjusted for these differences in the multivariable analysis to reduce any potential bias. As a result, we expect the outcomes to be less influenced by the baseline unbalance. Introduction Please give more specific main outcome that aimed to compare the short-term outcome of these 2 MIS procedures in this part Author Response: The Introduction section has been amended to recite more specific short-term outcomes compared. Methods Please give more detail about the stat used in this study. What about the patients aged equal to 60 Author Response: Statistical analysis section is amended to include more details. “60+” is revised to “≥ 60”. Please give more detail about the other comorbidity disease in these 2 group of study patients other than DM Author Response: In our study, we comprehensively assessed and considered various comorbidities in addition to DM in both groups, such as CKD and peripheral vascular disease. The distribution and detailed characteristics of these comorbidities across the two groups are presented in Table 1. Please give more detail about the surgical procedure other than hysterectomy such as pelvic and/or paraaortic node dissection Author Response: We agree with the reviewer regarding the importance of procedures like pelvic/paraaortic node dissection in this context. Regrettably, information of these specific procedures was not available in the NIS database, hence could not be analyzed. No detail about the preop prophylaxis antibiotic in these 2 MIS procedures Author Response: Data of prophylactic antibiotics usage were not available in the dataset, either. We have included this as one of the study limitations. What was the definition of obesity in this study? Author Response: The definition of obesity is BMI ≥ 30kg/m2. The description has been added. Results Why exclude the patients with lymph node invasion? Please add this exclusion criterion in the method part Author Response: The exclusion of patients with lymph node invasion was implemented to increase homogeneity in baseline characteristics of the study population. This criterion is added into Method. Page 6: The authors mentioned about after adjusted for relevant confounder …, please give more detail about the variable factors Author Response: The descriptions have been amended to include more details: “After adjusting for relevant confounders, including age, race, insurance status, household income, smoking, obesity, study year, hospital region, hospital location/teaching status, CKD, peripheral vascular disease, diabetes with chronic complications, and emergency admission” Discussion Summary of the main results Due to the unbalanced basic characteristics of these 2 participants in the robotic and laparoscopic procedure and the unclear stat that is used in this study, the result might be biased. Author Response: We have carefully adjusted for the differences in variables to reduce potential bias. Nevertheless, it is true that unmeasured confounders might exist and bias the results. We have added this issue into limitation section. Table 1: please specify the stat used to compare in this table (Chi-squared???) Author Response: The statistical methods employed are clarified in 2.6 Statistical Analysis in the article. Table 3: please give more detail about the variables that were adjusted. Which statistic was used? Author Response: Descriptions about adjusted variable were written in Table 3’s footnote, and more detail descriptions are added into the text. Table 4: What about the patients aged equal to 60? Author Response: Thanks for your comment. The '60+' group includes individuals aged 60 and older. We modified the description to '≥60' for clarity throughout the manuscript. Minor editing of English language required Author Response: The language of the manuscript has been edited and improved.Round 2
Reviewer 1 Report
Comments and Suggestions for Authors
Authors responsed to my comments.
Author Response
Author response: Thank you very much for your comment.
Reviewer 2 Report
Comments and Suggestions for Authors
Good job
Author Response

(The authors gave the same response as above.)
